# Inflammatory Bowel Disease Increases the Severity of Myocardial Infarction after Acute Ischemia–Reperfusion Injury in Mice

**DOI:** 10.3390/biomedicines11112945

**Published:** 2023-11-01

**Authors:** Wael Mami, Soumaya Znaidi-Marzouki, Raoudha Doghri, Melika Ben Ahmed, Sadri Znaidi, Erij Messadi

**Affiliations:** 1Plateforme de Physiologie et Physiopathologie Cardiovasculaires (P2C), Laboratoire des Biomolécules, Venins et Applications Théranostiques (LR20IPT01), Institut Pasteur de Tunis, Université Tunis El Manar, Tunis 1068, Tunisia; wael.mami@pasteur.utm.tn; 2Laboratoire de Transmission, Contrôle et Immunobiologie des Infections (LR16IPT02), Institut Pasteur de Tunis, Université Tunis El Manar, Tunis 1068, Tunisia; soumaya.marzouki@pasteur.tn (S.Z.-M.); melika.benahmed@pasteur.tn (M.B.A.); 3Département d’Anatomie et Cytologie Pathologiques, Institut Salah-Azaeiz, Université El-Manar, Tunis 1006, Tunisia; raoudha.doghri@gmail.com; 4Laboratoire de Microbiologie Moléculaire, Vaccinologie et Développement Biotechnologique (LR16IPT01), Institut Pasteur de Tunis, Université Tunis El Manar, Tunis 1068, Tunisia; sadri.znaidi@pasteur.utm.tn; 5Unité Biologie et Pathogénicité Fongiques, Département Mycologie, Institut Pasteur, INRA, 75015 Paris, France

**Keywords:** myocardial infarction, ischemia–reperfusion injury, inflammatory bowel disease, cytokines, animal model, translational medicine

## Abstract

(1) Background: Increased risk of myocardial infarction (MI) has been linked to several inflammatory conditions, including inflammatory bowel disease (IBD). However, the relationship between IBD and MI remains unclear. Here, we implemented an original mouse model combining IBD and MI to determine IBD’s impact on MI severity and the link between the two diseases. (2) Methods: An IBD model was established by dextran sulfate sodium (DSS) administration in drinking water, alone or with oral *C. albicans* (*Ca*) gavage. IBD severity was assessed by clinical/histological scores and intestinal/systemic inflammatory biomarker measurement. Mice were subjected to myocardial ischemia–reperfusion (IR), and MI severity was assessed by quantifying infarct size (IS) and serum cardiac troponin I (cTnI) levels. (3) Results: IBD mice exhibited elevated fecal lipocalin 2 (Lcn2) and IL-6 levels. DSS mice exhibited almost two-fold increase in IS compared to controls, with serum cTnI levels strongly correlated with IS. *Ca* inoculation tended to worsen DSS-induced systemic inflammation and IR injury, an observation which is not statistically significant. (4) Conclusions: This is the first proof-of-concept study demonstrating the impact of IBD on MI severity and suggesting mechanistic aspects involved in the IBD–MI connection. Our findings could pave the way for MI therapeutic approaches based on identified IBD-induced inflammatory mediators.

## 1. Introduction

Of all of the causes of cardiovascular disease (CVD), ischemic heart disease (IHD) remains the major cause of mortality and morbidity worldwide, despite significant advances in medical technologies in the diagnosis and treatment of the disease [1,2,3]. CVD may arise for various reasons including the steadily increasing incidence of obesity, type 2 diabetes, and genetic, environmental, dietary, and lifestyle factors. Besides all these, emerging evidence has shown that chronic inflammatory disorders, including rheumatic arthritis, systemic lupus erythematosus, and ankylosing spondylitis, are associated with an increased risk of heart disease and atherosclerosis [4,5,6,7]. At the same time, the relationship between inflammatory bowel disease (IBD) and CVD remains to be fully elucidated [8,9,10].

IBD is the most common systemic inflammatory disease with increasing incidence worldwide [11,12,13,14]. The disease comprises two major subtypes: ulcerative colitis (UC) and Crohn’s disease (CD) [15], characterized by chronic inflammation of the gastrointestinal (GI) tract with symptoms including diarrhea, blood and pus in stools, abdominal pain, fever, and weight loss. In UC, inflammation is mostly limited to the colon [16], while in CD, inflammation and fibrosis occur as patchy lesions throughout the GI tract [17]. IBD etiology includes various factors and is precipitated by environmental and genetic susceptibility [18,19]. Recent data also point to an intimate link between IBD and the gut-colonizing microbiota [15,20,21,22]. Because the host immune system is capable of tolerating the intestinal microbiota, while rapidly recognizing and destroying invading pathogens, disruption of the gut microbial equilibrium (i.e., dysbiosis) could evolve into an uncontrolled inflammatory response. In a key report, Sokol et al. showed that the fungal microbiota is altered in IBD patients [23]. This study particularly revealed that patients displayed increased GI colonization with the fungus *Candida albicans* (*C. albicans*) compared to healthy controls and indicated a deleterious link between *C. albicans* outgrowth and IBD pathogenesis.

Although IBD is associated with venous vascular events such as venous thrombosis [24,25,26,27,28], the extent of risk for IBD patients to develop coronary artery disease (CAD) and myocardial infarction (MI) in particular is not well understood. Considerable heterogeneity arises from cohort/meta-analysis studies of the association between IBD and increased risk of MI, and results have not been consistent across all reports. Several studies reported an elevated risk of MI in patients with IBD [29,30,31,32], while others reported no link [33,34] or even decreased hospitalizations after acute MI [35]. Possible mechanisms underlying the increased risk for cardiovascular disorders in patients with IBD have been proposed such as increased levels of inflammatory cytokines (i.e., tumor necrosis factor-α (TNF-α), interleukin-1β, (IL-1β), IL-6), oxidative stress, hypercoagulability, decreased numbers of circulating endothelial progenitor cells, and endothelial dysfunction [9,25,36,37,38].

Our proposed study is the first (i) to investigate whether a cause–effect relationship exists between IBD and the severity of MI, (ii) to precisely define the consequences of an inflammatory intestinal disease condition on the onset/development of CAD, and (iii) to assess the underlying mechanisms between the two diseases, using animal experimentation as a relevant approach when clinical hypotheses are discordant among patients [29,30,31,33,34,35]. To do this, we have for the first time set up an original experimental model by combining two distinct complementary and cross-validating in vivo animal models reproducing the context of IBD–MI in humans. The mouse IBD model of chemically induced colitis was set up by oral administration of dextran sulfate sodium (DSS) in drinking water, alone or with oral gavage with *C. albicans*, then validated by clinical and histological scores, as well as by intestinal and systemic inflammation assessment. MI was evaluated after coronary ligation-induced ischemia–reperfusion (IR) in mice by measuring infarct size (IS) and levels of cardiac troponin I (cTnI) in mouse blood, after 3 or 24 h of reperfusion, respectively [39,40,41,42,43,44].

Our work is the first proof-of-concept study to demonstrate the contribution of IBD to the severity of IR and to experimentally propose mechanistic aspects involved in MI susceptibility in IBD subjects. This could pave the way for preventive treatments against CVD in patients with IBD and provide a therapeutic approach for the treatment of MI in the context of IBD based on identified IBD-induced inflammatory mediators.

## 2. Methods

The experimental design for the study is shown in Figure 1 and Figure 2.

### 2.1. Animals

Male C57BL/6 mice (4–6 weeks old, 15–20 g) purchased from Janvier LAB (Janvier Labs, Le Genest-Saint-Isle, France) were housed under standard specific pathogen-free (SPF) laboratory conditions (temperature, 24 ± 1 °C; relative humidity, 50–60%; light cycle, 12/12 h light/dark), with unlimited access to controlled standard mouse chow and water. After two weeks of acclimation, mice were randomly divided into four groups to subsequently receive different treatments (Figure 1). The study was performed on 12-to-15-week-old male mice.

All animal experimental procedures in this study were performed in accordance with the Institut Pasteur de Tunis Ethical Committee (Permit No. 2017/12/LR11IPT08) and the recommendations of the Guide for the Care and Use of Laboratory Animals of the National Institutes of Health (NIH Pub. No. 85–23, Revised 1996).

### 2.2. Experimental Design of the Study

Since IBD is characterized by a succession of periods of flares and remissions, and it is during IBD flares that the cardiovascular event rate ratios are significantly increased [45], a model of acute colitis has proven to be most relevant for studying the cause–effect relationship and the underlying mechanisms of the IBD–IR connection. Hence, we set up an acute murine model of DSS-induced colitis, with or without *C. albicans* oral gavage (Figure 1), which was shown to mimic UC in human [46]. DSS concentration and treatment time were determined in a preliminary pilot study to allow efficient induction and maintenance of intestinal inflammation, without significantly altering the general status of the animals, thus they could be subsequently subjected to IR experiments (Figure 2).

After acclimation, mice were randomly divided into four groups: control group (Ctrl), dextran sulfate sodium (DSS) group, *C. albicans* (*Ca*) group, and DSS + *Ca* group (Figure 2). Experimental groups were composed of 4 to 8 animals. Mice in the control group were given sterile water during the study period (17 days). From day 0, mice in the DSS groups were given 1% (*w*/*v*) DSS (molecular weight: 36–50 kDa, MP Biomedicals, Santa Ana, CA, USA) ad libitum in drinking water for 10 days (0 to 9) to induce colitis, and then DSS was withdrawn and replaced with water to allow remission until day 17. In the experiments that included *C. albicans* treatment (*Ca* and DSS + *Ca* groups), mice were orally gavaged, on day 0, with 200 µL of phosphate-buffered saline (PBS) suspension containing 5 × 10^7^ *C. albicans* SC5314 live cells. Mice in the control group and receiving only DSS were given 200 μL sterile PBS by oral gavage on day 0 instead of the *C. albicans* suspension.

Stool samples were collected every two days starting from day 1 post-gavage (day 0 was used as negative control before gavage treatment). The presence of yeast in the GI tract was followed, through plate counts of fungal colonies in stool suspensions collected from each animal. *C. albicans* colonies were counted after 48 h of incubation at 30 °C on yeast peptone dextrose (YPD) agar medium supplemented with 1 g/L chloramphenicol and 50 mg/l gentamycin to prevent bacterial growth and the results were expressed as colony-forming units per gram of feces (CFU/g).

On day 17, the animals were either sacrificed by cervical dislocation, and their blood collected beforehand by submandibular venipuncture, or subjected to IR (30 min of coronary occlusion followed by 180 min reperfusion for IS measurement or 24 h reperfusion for cTnI serum level assessment). Serum samples were stored at −80 °C until use. The entire colon from cecum to anus was removed, colon length was measured, and various anatomical sections were stored in 10% buffered formalin until use for histological analysis.

### 2.3. Inflammatory Bowel Disease Studies

#### 2.3.1. *Candida albicans* Strain SC5314 Preparation

The *Candida albicans* SC5314 reference strain [47] was cultured in 10 mL liquid YPD medium overnight at 30 °C with continuous shaking (150 rpm) using an INFORS HT Ecotron AG CH-4103 shaker/incubator (Infors AG, Bottmingen-Basel, Switzerland). Cells were then collected by centrifugation at 1500 rpm for 5 min. The supernatant was discarded and the cell pellet was washed twice by adding 10 mL of PBS followed by centrifugation at 1500 rpm for 5 min. The yeast pellet was suspended in 5 mL of PBS and the optical density (OD) at 600 nm (OD_600nm_) was measured. Gavage solution was prepared based on 1 OD_600nm_ = 10^7^ cells/mL (25 × 10^7^ cells/mL for 200 µL gavage volume/mouse).

#### 2.3.2. Disease Activity Index Assessment

All mice were monitored daily for body weight and fecal conditions following the initiation of DSS treatment. A disease activity index (DAI) was recorded based on body weight loss, fecal consistency, and hematochezia (Table 1). The DAI score (0–10) was the total score of the above three parameters [48,49,50] (Table 1). For analyzing DAI, areas under the curve (AUCs) were calculated within each experimental group using GraphPad Prism (version 8.0.2 GraphPad Software, San Diego, CA, USA).

#### 2.3.3. Histological Analysis

After mouse sacrifice, dissection was performed and the GI tract was extracted and unrolled. The length of the colon, from cecum to rectum, was measured and sections of the ends of the colons were collected in separate clean tubes and snap-frozen in liquid nitrogen for molecular studies. The rest of the organs of the GI tract were fixed in 10% buffered formalin and embedded in paraffin for histological studies. Samples were sliced into 5 µm sections for periodic acid–Schiff (PAS) staining to determine the presence of *C. albicans* cells, by using microscopy or hematoxylin and eosin (H&E) staining to assess inflammatory lesions according to standard protocols [51]. Randomly selected slices from each group were observed and analyzed by a blinded pathologist using an Olympus BX series multihead microscope equipped with a DP21 camera (Olympus Life-Science, Shinjuku, Tokyo, Japan), at 40× magnification. A histological score calculated based on inflammatory cell infiltration (1–3 points) and the intestinal architecture (1–3 points) was used to assess the degree of the histopathological changes, as previously described [52] (Table 2). Focally increased numbers of inflammatory cells in the *lamina propria* were scored as 1, confluence of inflammatory cells extending into the submucosa as 2, and transmural extension of the infiltrate as 3. For tissue damage, discrete lymphoepithelial lesions were scored as 1, mucosal erosions as 2, and extensive mucosal damage and/or extension through deeper structures of the bowel wall as 3. The two equally weighted subscores (cell infiltration and tissue damage) were added and the combined histological colitis severity score ranged from 0 to 6.

#### 2.3.4. *Candida albicans* Gastrointestinal Colonization Analysis

The presence of yeast in the GI tract was followed every two days during the 17-day treatment through colony counts on YPD agar medium from samples collected from each animal. Stool samples were collected every two days in separate tubes, weighed, homogenized in 1 mL PBS, then serially diluted (1/10, 1/100, and 1/1000) for CFU counting. One hundred microliters of the suspension was plated onto solid YPD medium supplemented with 1 g/L chloramphenicol and 50 mg/L gentamycin prepared in 90 mm Petri dishes. *C. albicans* colonies were counted after 48 h incubation at 30 °C and the results expressed as colony-forming units (CFUs) per gram of stool (CFU/g).

#### 2.3.5. Fecal Lipocalin 2 Analysis

Stool samples were collected every two days during the 17-day treatment, weighed, and then solubilized in PBS supplemented with 0.1% Tween 20 at a concentration of 100 mg/mL. Samples were then vortexed for 20 min and the supernatant was collected and frozen at −20 °C following centrifugation at 12,000 rpm for 10 min at +4 °C until use as previously described [53]. To determine the amount of Lcn2, we have performed ELISA using the Mouse Lipocalin-2/NGAL DuoSet ELISA kit (R&D systems, Minneapolis, MN, USA) following the manufacturer’s instructions. Briefly, plates were pre-coated with 100 µL of primary Lcn2 antibody (capture antibody) reconstituted in PBS following the manufacturer’s recommended concentration (480 µg/mL stock and 4 µg/mL working concentration in our case), sealed, and incubated overnight at room temperature (RT°). The following day, plates were washed 3 times with 400 µL of PBS containing 0.05% Tween 20 using a ROCKER BioWasher200 Microplate ELISA Washer (ROCKER Scientific, New Taipei City, Taiwan). Three hundred microliters of reagent diluent composed of 1% bovine serum albumin (BSA) in PBS was added to each well to saturate/block the plate and incubated for 1 h at RT followed by one washing step. One hundred microliters of 2× serial dilutions of Lcn2 standard was reconstituted and diluted in reagent diluent, following the manufacturer’s instructions (500 to 7.81 pg/mL in our case), and diluted samples were added to assigned wells and incubated for 2 h at RT followed by one washing step. One hundred microliters of biotinylated Lcn2 detection antibody reconstituted and diluted in reagent diluent (30 µg/mL stock and 500 ng/mL working concentration in our case) was added to each well and incubated for 2 h at RT followed by one washing step. One hundred microliters of 40-fold-diluted streptavidin–horseradish peroxidase (HRP) was added to each well and incubated for 20 min at RT protected from light followed by one washing step. One hundred microliters of tetramethylbenzidine (TMB) and H_2_O_2_ 1:1 mixture substrate solution (TMB ELISA Substrate High Sensitivity, Abcam, Cambridge, UK) was added to each well and incubated for 20 min at RT protected from light. Fifty microliters of 2N sulfuric acid (H_2_SO_4_) stop solution was added to each plate and optical density (OD) of each well was determined immediately using a TECAN SPARK multiplate reader (Tecan Group Ltd, Männedorf, Switzerland) at 450 nm wavelength with 540 nm correction.

#### 2.3.6. Serum Cytokine Analysis

To assess the extent of inflammation outside the GI tract, blood samples were collected from mice via the facial vein using a Goldenrod animal lancet (Medipoint, Mineola, NY, USA) on days 0, 9, 13, and 17. Blood was left to clot for 1 h, then serum was collected after centrifugation at 3500 g for 10 min at 4 °C. Blood samples were stored at −80 °C until use and were thawed only once. Cytokines, interleukin-2 (IL-2), IL-4, IL-6, IL-10, IL-17A, TNF-α, and IFNγ in the serum were quantified using an Aimplex Mouse Th1/Th2/Th17 7-Plex Panel bead-based immunoassay kit (AimPlex Biosciences, Pomona, CA, USA) following the manufacturer’s instructions. Briefly, 45 µL of antibody-coupled bead suspension mix was added to each well and buffer was removed by vacuum aspiration. Then, 45 µL of diluted standard mix as well as 15 µL samples were added to assigned wells prefilled with 30 µL of rodent serum/plasma/body fluids (SPB) assay buffer. Plates were sealed and incubated for 60 min on a plate shaker at 700 rpm. Then, fluids were removed by vacuum aspiration and the wells were washed 3 times by adding 100 µL of wash buffer. Twenty-five microliters of biotinylated antibodies were added to each well and the plate was incubated for another 30 min under the same conditions. After washing, 25 µL of streptavidin–phycoerythrin (PE) was added to each well and incubated for 20 min, then washed twice. Beads were then resuspended in 300 µL of reading buffer and were collected in FACS tubes and analyzed with a BD FACSCanto II flow cytometer (BD Biosciences, Franklin Lakes, NJ, USA).

### 2.4. Ischemia–Reperfusion Studies In Vivo

#### 2.4.1. Surgical Preparation and Induction of Myocardial Infarction

On day 17, mice were anesthetized with an intraperitoneal sodium pentobarbital injection (60 mg/kg, Ceva Santé Animale, Libourne, France). The animals were intubated and ventilated (200 µL/breath at a rate of 170 breaths/min), using a MiniVent Type 845 mouse ventilator (Hugo Sachs Elektronik, Harvard Apparatus, March-Hugstetten, Germany). A catheter was inserted into the jugular vein. Body temperature was monitored with a HB101/2 homoeothermic system temperature control unit (Panlab technology for bioresearch, Barcelona, Spain) via a connected rectal probe and maintained at 37 °C using a heating pad. The electrocardiogram (ECG) and heart rate (HR) were recorded throughout the experiments using under-skin probes connected to EmkaIox2 software (Emka Technologies, Paris, France). A left thoracotomy was performed to expose the heart, and the pericardium was removed. The left anterior descending coronary artery was occluded with an Ethicon Prolene 8.0 polypropylene suture (Johnson & Johnson, New Brunswick, NJ, USA) 2 mm from the tip of the left atrium for 30 min. Successful coronary occlusion was verified by the development of a pale color in the distal myocardium and by ST segment elevation and QRS widening on the ECG. After 30 min of sustained ischemia, coronary blood flow was restored by loosening the suture. Successful reperfusion was confirmed by visualization of hyperemic response and restoration of normal ECG. The lungs were then reinflated by increasing positive end expiratory pressure, and the chest was closed. Reperfusion was maintained for a 3 h period [39,40,41,42].

#### 2.4.2. Infarct Size Measurement

After reperfusion, the chest was reopened, the coronary artery was reoccluded, and 0.5 mL of 5% Evans blue (Sigma-Aldrich, Merck KGaA, Darmstadt, Germany) in PBS solution was injected as a bolus into the jugular vein in order to delineate the area at risk (AAR), which remained unstained by the Evans blue. The heart was excised, and the left ventricle (LV) was isolated, weighed, and sliced into 4 transverse pieces from base to apex, the first cutter blade being positioned at the site of the coronary occlusion. The slices were weighed, and color digital images of both sides of each slice were obtained with an EOS 550D camera (Canon, Tokyo, Japan) connected to a Leica M80 stereomicroscope (Leica Microsystems, Rueil-Malmaison, France). The slices were then incubated at 37 °C with buffered 1% 2,3,5-triphenyltetrazolium chloride (TTC) (Sigma-Aldrich, Merck KGaA, Darmstadt, Germany) solution for 20 min. Viable myocardium, which contained dehydrogenases, reacted with TTC and was stained brick red, whereas any necrotic tissue remained unstained due to the lack of active enzymes. The tissue sections were then fixed in a buffered 10% formalin solution for 24 h before being photographed again to delineate the IS [39,40,41,42,43]. The cross-sectional area, the lumen area, the AAR (unstained by Evans blue), and IS (unstained by TTC) of the LV were outlined on each color image and quantified by a masked observer using a computerized planimetric technique (ImageJ software, NIH, Bethesda, MD, USA). The absolute weights of AAR and IS were then calculated for each slice. The sum of the absolute weight values of AAR and IS of the 3 ischemic slices of each heart was calculated and expressed as a percentage of the total weight of the slice. The ratio of IS to AAR was calculated from these absolute weight evaluations and expressed as a percentage of AAR.

#### 2.4.3. Serum Cardiac Troponin I (cTnI) Determination

As an additional readout for myocardial infarct severity, we measured cTnI levels in circulating mouse blood, with or without DSS treatment or *C. albicans* gavage, after a 24 h reperfusion period so that IR lesions were full blown at this time (*n* = 5–8/group). Blood samples were collected from the submandibular vein at the end of 24 h reperfusion. Serum cTnI levels were determined with a mouse cardiac quantitative cTnI assay (Life Diagnostics, West Chester, PA, USA) [54]. Briefly, 100 µL of sera diluted 1:10 to 1:100 along with a serially diluted standard were added to troponin-antibody-pre-coated plate wells and sealed. The incubation was carried out with continuous shaking at 150 rpm and 25 °C for 2 h. Plates were then washed 5 times using the provided wash buffer by adding 400 µL each time using our ROCKER plate washer. Then, 100 µL of diluent as well as 100 µL of streptavidin–HRP conjugate were added to each well and the plate was incubated for 1 h as described above. After repeating the washing procedure, 100 µL of TMB substrate was added to each well, incubated for 20 min, then 100 µL of acid stop solution was added to each well. Absorbance was then determined within 5 min using the TECAN SPARK plate reader at a wavelength of 450 nm.

### 2.5. Cardiomyocyte Studies In Vitro

Although high-molecular-weight DSS has been reported not to be absorbed into the bloodstream [55], in vitro cardiomyocytes have been used to test the effect of DSS per se on cardiac cells and determine if it was itself inducing potential heart damage if it eventually passed into the bloodstream.

Human induced pluripotent stem cells (HiPSCs) were differentiated by cytokine induction into contracting cardiomyocytes (HiPSC-CMs). The cells were seeded in a 24-well plate pre-coated with a matrix at a density of 1 × 10^5^ to 1.5 × 10^5^ cells/well and incubated for 24 h to adhere and regain contraction. The culture medium was then replaced by another supplemented with DSS at different concentrations (15, 30, and 50 µg/mL). The plate was incubated for 48 h, then supernatants were collected into annotated tubes and the wells were treated with TryPLE enzyme mix to digest the matrix and dissociate the cells. Each suspension was added to its respective supernatant tube and centrifuged at 1200 RPM for 5 min. Cell pellets were washed twice with PBS and then resuspended in 200 μL of Muse Count & Viability kit solution (LUMINEX, Luminex Corporation, Austin, TX, USA). The percentage of viable cells was determined by flow cytometry using LUMINEX’s Guava Muse Cell Analyzer (Luminex Corporation, Austin, TX, USA).

### 2.6. Statistical Analysis

Data were presented as mean ± SEM. To determine significance among the four groups, the non-parametrical Mann–Whitney test was used, and one-way ANOVA was followed by Student’s *t*-test for further evaluation of differences between two means. All statistical analyses and graph generation were performed by using GraphPad Prism (version 8.0.2 GraphPad Software, San Diego, CA, USA). *p* < 0.05 was considered to be statistically significant.

## 3. Results

### 3.1. Disease Activity Index Evolution in IBD Mice

AI was recorded based on body weight loss, fecal consistency, and hematochezia (see Materials and Methods). The DAI showed that DSS treatment induced a weight loss of up to 18% (day 9) as well as a loss of stool consistency and the appearance of colorectal bleeding. Disease severity was at its maximum on day 9 of DSS administration (7.9 ± 0.7; *p* < 0.001 vs. DSS group on day 0, Figure 3A), where mice displayed diarrhea with rectal bleeding and significant weight loss with reduced mobility as well as food and water intake. *C. albicans* gavage treatment, alone or in combination with DSS, had no impact on DAI as evidenced by AUCs (Figure 3A,B).

After DSS removal, the general condition of the animals improved concomitantly with a regain of weight, a reduction in bleeding, and solidification of the stools, but always with the presence of traces of blood. On day 17 (i.e., 8 days after DSS removal), the DAI score was significantly reduced compared to day 9 (2.6 ± 0.6; *p* < 0.01, Figure 3C), with the resumption of activity and weight of mice and an improvement in their general state of health, close to normal. A few traces of blood and pasty stools were detectable at this stage.

### 3.2. Histological Analysis

To establish a histological score, each organ of the GI tract was scored separately (Figure 4A) and the overall histological score (Figure 4B) showed a significant progressive inflammation increase in animals treated with *Ca* alone, DSS alone, and a combination of DSS + *Ca* (i.e., *Ca* < DSS < DSS + *Ca*) compared to the control group (Figure 4B). In DSS-treated mice, histological study of sections of the GI tract showed severe infiltration of inflammatory neutrophils into the mucosa and submucosa of the colon with ulceration and loss of crypt architecture as well as focal lesions (Figure 4C), reflecting the histological score assessment (DSS group: 7.3 ± 1.21, *p* < 0.001 vs. control group, Figure 4B). This effect was accompanied by colon shortening in the DSS-treated groups (Figure 4D).

In mice receiving *C. albicans* gavage, histological abnormalities were observed only in the stomach (Figure 4A), which may indicate a potential role of *C. albicans* in the progression and severity of inflammation particularly in this section of the gut, while *C. albicans* had no additional impact on inflammation when combined with DSS treatment (10.1 ± 1.68; *p* < 0.001 vs. DSS group, Figure 4A).

### 3.3. Candida albicans Colonization Assessment

After *C. albicans* oral gavage on day 0, fungal colonization was readily detected on day 1 post-gavage and persisted throughout the treatment period (Figure 5A), varying from 5 × 10^5^ up to 10^6^ CFU/g. Colonies were still present on the day of animal sacrifice (day 17 after gavage, Figure 5A) in both *Ca*-treated and DSS + *Ca*-treated mice, indicating a stable *C. albicans* colonization of the GI tract of mice that underwent the subsequent IR experiments. PAS-stained slices from post-mortem study of GI tract organs showed invasions and colonization of the stomach and colon walls by *C. albicans* with the presence of morphologies ranging from yeast to true hyphae (Figure 5B). *Ca* mice showed several ulcerative foci at the periphery of colonies suggesting possible *Ca*-induced inflammation.

### 3.4. Local Inflammatory Marker Analysis

Lcn2 levels showed a marked increase starting on day 5 of DSS administration, reaching levels as high as 336.5 ng/mL on day 9 (day 0: 1.5 ± 0.7 vs. 336.6 ± 124.05 ng/mL on day 9, *p* < 0.001, Figure 6). Although DSS treatment was discontinued on day 9, Lcn2 levels remained relatively high and displayed steady state levels from day 9 up to day 17 (day 17: 505.5 ± 129.8 ng/mL), indicating a persistent inflammatory state in mice following removal of DSS (Figure 6). These levels showed that while animals expressed a general healing process objectified by the DAI decrease (Figure 3), local inflammatory markers such as Lcn2 persisted at high levels.

### 3.5. Circulatory Cytokine Analysis

Out of the seven inflammatory cytokines tested, only IL-6 (Figure 7A,B) displayed high expression and a significant increase in the groups treated with DSS on day 9 (314.2 ± 53.9 ng/mL) compared to control animals (14.17 ± 0.9 ng/mL, *p* < 0.001) and *Ca* groups (16.42 ± 2.3 ng/mL, *p* < 0.001). IL-6 levels remained significantly high following DSS removal compared to the control group and until the end of treatment on day 17, when the animals underwent the IR maneuver (170.1 ± 90 ng/mL, *p* < 0.01, Figure 7A). *C. albicans* gavage had no impact on IL-6 release alone or in combination with DSS (Figure 7A).

The remaining pro-inflammatory cytokines (TNF-α, IFNγ, and IL-17) displayed similar expression dynamics, with values not exceeding 10 pg/mL (Figure 7C–E). No significant changes were observed for anti-inflammatory cytokine IL-4 (Figure 7F), while IL-10 was not detectable in any of the experimental conditions.

### 3.6. Infarct Size Quantification

There was no difference in body weight (BW), left ventricle (LV) weight, AAR to LV ratio, or HR before occlusion of the coronary artery or after reperfusion between the different experimental groups, regardless of whether they had been subjected to treatment or IR (Table 3, Figure 8A,B). After IR, the IBD mice developed a significantly more severe infarct injury (Figure 8C,D) (IS/AAR: 32.6 ± 3.3%, *p* < 0.01) than the control mice (19.7 ± 2.5%). *C. albicans* inoculation of mice did not significantly increase IS when alone (23.9 ± 3.0%) or in combination with DSS (38.5 ± 3.3%) (Figure 8D).

### 3.7. Serum Cardiac Troponin I Quantification

Correlating with IS increase, serum cTnI levels in DSS-treated mice have been found to be significantly higher than in controls (51.17 ± 10.5 ng/mL versus 13.14 ± 3.6 ng/mL, *p* < 0.05) after 24 h of reperfusion (Figure 8E). Fungal colonization increased cTnI in the DSS + *Ca* group (105.46 ± 33.7 ng/mL), but this increase was not significant compared to the DSS group (Figure 8E). Correlation between IS/AAR and serum cTnI was significant (r^2^, *p* < 0.05).

### 3.8. DSS Effect on Cardiomyocytes In Vitro

DSS had no effect on HiPSC-derived cardiomyocyte viability (Figure 9).

Although DSS has been shown not to be metabolized in the GI tract [55,56], our finding indicates that the aggravation of IS in the DSS-treated groups may be due to the passage into the blood of cytokines or intestinal microflora products in the DSS-induced colitis model.

## 4. Discussion

CVDs are the most frequent cause of death worldwide. Their prevalence is higher in several chronic inflammatory conditions. This is likely due to the substantial contribution of the inflammatory status of the patients to the increased cardiovascular risk [4,5,6,7,57]. IBD is characterized by chronic, progressive inflammation of the GI tract and elevated pro-inflammatory markers. However, the link between IBD and increased risk of MI is not well defined yet. Despite large database and observational meta-analysis studies [32,34], the association between IBD and MI remains controversial [29,30,31,33,34,35,58]. To date, only increased risk of venous thrombotic events has been significantly reported in patients with IBD, whereas the risk of arterial diseases (such as MI) has been less well characterized [59,60,61,62]. We sought to assess this risk and to identify potential pathogenic mechanisms using animal experimentation by setting up, for the first time, an original animal model reproducing in vivo the context of IBD–MI in humans. Due to the remarkable increase in the rate ratios of cardiovascular events during IBD flares [45,58], an acute colitis model with a 9-day DSS treatment phase was carried out in mice to develop an IBD that morphologically and symptomatically resembles UC in humans [50]. DSS is used in mice to induce colitis by causing erosion of epithelial cells and increasing colonic mucosal permeability, leading to hyperemia, ulcerations, submucosal edema, and histological changes, including infiltration of granulocytes [50]. Classically, 1.5% to 5% DSS in animal drinking water is used for a period ranging from 5 to 8 days to induce acute colitis [50]. However, in our model, we administered DSS orally at a low dose (1%) so that IBD severity could be consistent with subsequent myocardial IR experiments. Indeed, a more severe IBD would not have been compatible with animal survival after cardiac experiments, as found in a preliminary pilot study. In addition, animals were allowed an 8-day remission period where DSS was replaced with water for better animal survival in IR experiments. However, at the time of IR (day 17), we ensured that blood cytokine levels and inflammatory status of the animals were comparable to those measured during the peak of IBD on day 9, which allowed us to reliably study the effect of IBD on the severity of ischemia (Figure 10).

Our results showed that DSS treatment, which did not induce any deleterious effect per se on cardiomyocytes in vitro, aggravated the severity of IR by inducing a two-fold increase in IS compared to controls, which was also reproduced by cTnI quantification assay (Figure 8E). AAR, HR, and cardiac morphological parameters were similar in all the experimental groups (Figure 8, Table 3), suggesting that there were no anatomical differences in the coronary vessels or left ventricles among these mice and no major differences in their basal cardiac metabolism that could have influenced the results of IR and that the differences in IS are only due to the treatment imposed on mice (DSS/*Ca*). The IS values, which averaged ~20% under basal conditions in the present study, were very similar to those reported by others in mice subjected to 30 min of coronary occlusion followed by 180 min reperfusion [63]. Although TTC staining is the method of choice for post-mortem determination of IS [43,64], an additional readout for myocardial infarct severity was performed by measuring serum cTnI levels in the different experimental groups [40,65]. A strong correlation between IR (IS/AAR (%)) and cTnI levels in mice was found (Figure 8F, r^2^ = 0.96, *p* < 0.05).

In MI, abnormalities of endothelial function play an important role in the pathogenesis of CAD leading to decreased endothelium-dependent vascular relaxation, impaired myocardial perfusion, and thrombotic events [66,67]. Being a chronic inflammatory disease, IBD is associated with an upregulation of several cytokines [68]. Due to the disruption of the intestinal mucosal barrier during IBD, it has been suggested that a possible mechanism underlying the increased MI severity in IBD includes the translocation of microbial lipopolysaccharides (LPSs) that can stimulate the production of pro-inflammatory molecules leading to endothelial injury and atherosclerosis [9]. Some of them, such as TNF-α and C-reactive protein, are known to be mediators of atherosclerosis and endothelial alterations by damaging vascular functionality and reducing the nitric oxide (NO) availability [9,37,38,69,70,71]. By quantifying systemic cytokines in IBD mice, we detected an increase in the pro-inflammatory cytokines TNF-α, IFNγ, and IL-17, especially on day 17 and in the DSS + *Ca* group. Of all these cytokines, only IL-6 was found to be significantly expressed in the blood and increased in the DSS-induced IBD groups, with levels remaining elevated on day 17 when the animals were sacrificed (Figure 10). Previous studies have shown that IL-6, whose levels are increased in IBD [72,73,74,75], is linked to endothelial dysfunction, thrombocytosis, early atherosclerosis, and CAD [76,77,78,79]. IL-6 is a key risk factor for CVD by triggering inflammatory reaction and inducing other molecules’ release [80,81,82]. These observations could explain why, in our work, IBD animals displaying high levels of IL-6 developed greater susceptibility to ischemia by developing more severe infarct after IR. Our results are in line with studies reporting that a higher IL-6 level measured 24 h after MI is associated with larger IS and diminished cardiac function [83]. Clinical data have also shown that levels of IL-6 can serve as a biomarker of mortality in CAD [84], with a strong relationship to future cardiac events and mortality in healed MI patients [85]. It should be noted that IL-6 plays the same role in IBD as in CAD, as a major factor in the pathogenesis of IBD [86], and that inhibition of IL-6 prevents development of both CD [87,88] and IHD, even in patients refractory to conventional drugs such as corticosteroids [88]. These results suggest shared inflammatory mechanisms between MI and IBD. MI, especially followed by reperfusion, leads to a complex post-ischemic inflammatory response resulting from the death of myocardial cells by necrosis of the infarcted tissues [43,89]. Infiltration of the ischemic site by immune cells can both promote cardiomyocyte death, inflammation, and subsequent post-MI remodeling and heart failure, as well as facilitate regeneration of damaged heart muscle at a later stage [90,91,92,93]. At the onset of MI, necrotic myocytes release damage-associated molecular patterns (DAMPs) and cytokines that activate the recruitment of non-specific immune cells such as monocytes, neutrophils, and dendritic cells from the peripheral vasculature into the infarcted area [94]. After the inflammatory phase, a specific immune response involving T and B lymphocyte infiltration initiates the repair of MI, in particular via the production of cytokines by regulatory T cells to promote macrophage polarization and myocardial healing [94]. Resolution of inflammation is achieved by macrophage recruitment in the injured site to polarize into anti-inflammatory macrophages that activate MI repair by secreting anti-inflammatory cytokines and eliminating necrotic cells [95]. Perturbations affecting both balance and transition between pro- and anti-inflammatory phases can exacerbate acute MI and thus aggravate post-infarction clinical outcomes [96,97]. In our study, the aggravation of MI in IBD mice could likely lead to an intense and unbalanced inflammatory reaction, affecting not only MI size but also post-ischemic remodeling and repair. Therefore, insight into immune cell-mediated inflammation in MI could contribute to the identification of appropriate and effective therapeutic targets to modulate macrophage polarization toward an anti-inflammatory phenotype [95] and improve the reparative phase following MI in IBD subjects.

Besides the inflammatory impact of IBD-induced cytokines on cardiac function, prolonged systemic inflammation can cause platelet aggregation and impaired coagulation [27,71] leading to deregulation of the coagulation system and to thrombus formation, which causes acute coronary syndrome in IBD patients [98]. Additionally, other factors may influence the risk of MI in IBD subjects, including certain products of the intestinal microflora which may enter into circulation through the damaged intestinal mucosal layer and activate inflammatory responses which could lead to endothelial dysfunction, arterial stiffness, and atherosclerosis [9,36,99,100]. In this context, several studies have reported that gut microbiota dysbiosis is linked not only to IBD but also to cardiovascular risk factors, such as hypertension, atherosclerosis, MI, heart failure, and diabetes [101]. More particularly, thrombus microbiota modifications could affect both the development and progression of atherosclerosis [102]. Recent reports have shown that thrombus colonization by specific bacterial genera such as *Prevotella* can increase coronary thrombus burden and plaque vulnerability to rupture [102,103]. These effects may represent the result of *Prevotella*’s actions in increasing trimethylamine-N-oxide (TMAO) levels and consequent effects on CDL40 and von Willebrand factor thrombus contents, as reported in experimental studies that have clearly emphasized the role of TMAO in increasing coagulation [104]. In these studies, authors found that *Prevotella* thrombus contents were significantly associated with inflammatory mediators such as neutrophil activation markers [103]. The GI microbial flora [105,106] also includes a gut mycobiota, among which *C. albicans* is one of the most prevalent human fungal pathogens [46,107]. Several data have pointed out the link between the outgrowth of *C. albicans* and the pathogenesis of IBD, since IBD patients display increased GI *Candida* colonization [23] and produce anti-yeast antibodies that correlate with the severity of IBD [108,109,110]. IBD-induced microbial dysbiosis and *Candida* overgrowth have also been shown to be associated with increased levels of pro-inflammatory mediators [46,111]. Unlike in humans, *Candida* is not prominent in the mouse gut, does not colonize the mouse GI tract, and is undetectable by culture methods [112,113]. Therefore, DSS colitis with *C. albicans* administration might be a more adequate representative model (DSS + *Ca*) to resemble the human condition [23]. Thus, a DSS colitis model with *Candida* administration was first investigated to replicate the IBD human condition, and then to test the influence of fungi on MI in the DSS model. *C. albicans* outgrowth, which has been reported to be promoted by DSS chemically induced colitis [114], has been shown to aggravate gut inflammation in these murine models [46,114]. However, in our experiments, mouse oral gavage with *C. albicans* on day 0 did not significantly worsen IS, neither in basal condition nor in combination with DSS. This suggests that, although *C. albicans* was detectable at the time of IR in the GI tract (on day 17), *C. albicans* has no effect on MI severity by itself in our experimental conditions. A single oral gavage of *C. albicans* after DSS administration in wild-type mice demonstrated little impact on the DSS mouse model, possibly due to a non-sustained or a subtle presentation of gut fungi [115]. Similarly, in our work, single oral inoculation of *C. albicans* failed to significantly increase the level of local inflammatory markers such as Lcn2 or circulating cytokines (Figure 6 and Figure 7). Hence, a DSS model with multiple *C. albicans* administrations would likely be required for a sustained and significant effect on inflammatory markers and subsequent MI aggravation. In the literature, models with several administrations of C*. albicans* demonstrated a higher impact on the DSS mouse model as well as an exacerbation of intestinal mucosal barrier disruption and colitis severity through leaky gut-enhanced systemic inflammation [116,117,118]. In these models, repeated *Candida* gavages were necessary because lower doses and less frequent gavages did not consistently induce *Candida* presentation in the gut [117].

Although our results provide novel insights into the interaction between IBD and MI, they leave some unanswered questions. Indeed, we demonstrated for the first time that IBD significantly aggravated the severity of MI and that these effects could involve IL-6 pathway activation. However, the role of IL-6 as a causal link between IBD and MI was not further explored in this work. Several reports have demonstrated that, in both IBD and MI, inhibition of IL-6 can prevent disease development, even in patients refractory to clinical trials with anti-TNF-α therapies [88,119,120]. Thus, IL-6 can be a potential biomarker for MI prognosis and a target to improve the treatment of IBD patients. In the future, further studies will be needed to assess the role of this cytokine in the predisposition to CVD in individuals with IBD. Furthermore, in our work, microbiota modification by acute inoculation of *C. albicans* had no impact on DSS-induced colitis, suggesting that multiple *C. albicans* administrations will likely be required for a higher impact on MI. Finally, further investigations will be needed to determine whether other gut microflora products, such as metabolites, are responsible for the severity of IS in the DSS-induced colitis model.

## 5. Conclusions

With more than 10 million patients worldwide, IBD has become a global concern, particularly for the deleterious intestinal and extra-intestinal consequences it induces, such as the increased risk of CVD [121]. This highlights the need for IBD prevention research and innovations in health care systems to manage this complex and costly disease. In this study, we have for the first time provided an answer to a discordant human hypothesis, particularly concerning the relationship between IBD and increased risk of MI, for which observational studies could not provide any direct evidence [29,30,31,33,34,35]. IBD-induced cytokines may illuminate novel pathways to disease development and serve as biomarkers that can be assessed using diagnostic tests for both susceptibility and severity of MI. By using animal experimentation and establishing an original mouse model combining both IBD and MI, we demonstrated for the first time that IBD significantly aggravated the severity of MI and that the causal link could involve IL-6 pathway activation. Although mouse experimental data should be extrapolated with caution to human disease, we believe that these findings, after crossing them with clinical approaches, could have a significant impact on the prevention and improvement of treatment/prognosis of acute CAD in IBD patients.

## Figures and Tables

**Figure 1 biomedicines-11-02945-f001:**
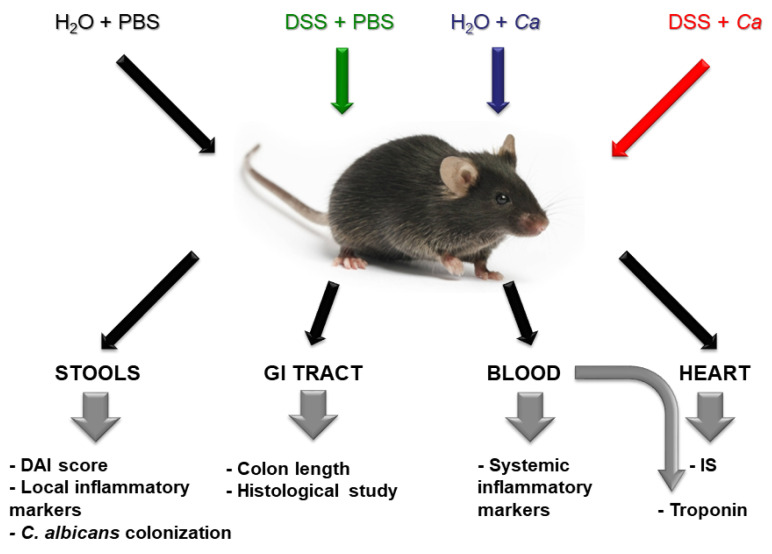
Protocol to examine the relationship between inflammatory bowel disease and severity of myocardial infarction. Chemically induced colitis in mice will be established by administration of DSS in drinking water. In addition, mice will be gavaged by *C. albicans* (*Ca*) to mimic intestinal dysbiosis in humans. Control group as well as *Ca* group will receive only water instead of DSS and control and DSS groups will be gavaged with vehicle PBS instead of *C. albicans* suspension. Stool samples and blood will be collected during treatment period to assess disease evolution. GI tract organs as well as the hearts will be harvested at the end of treatment for histological studies as well as IS measurement.

**Figure 2 biomedicines-11-02945-f002:**
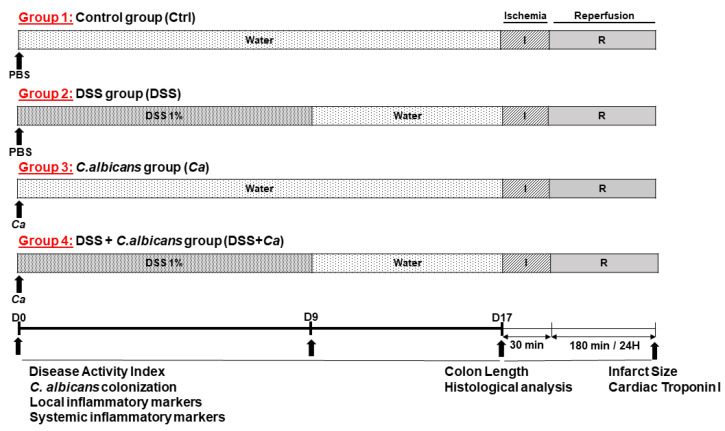
Experimental design of the study. Chemically induced inflammatory bowel disease (IBD) was performed by adding dextran sulfate sodium (DSS, 1% (*w*/*v*)) to drinking water from day 0 to day 9 to induce colitis, and DSS was removed on day 9 to allow remission until day 17 (DSS group). A second group of healthy mice, given only water, was used as control (Ctrl). A third group of mice was orally gavaged on day 0 with 200 µL of PBS containing 5 × 10^7^
*C. albicans* live cells without any other treatment (Ca). A fourth group was treated with DSS and orally gavaged with *C. albicans* (DSS + *Ca*). All animals underwent 30 min ischemia followed by either 180 min or 24 h reperfusion for IS or cTnI serum level determination, respectively. The presence of yeast in the intestinal tract was identified in the stool samples collected from each animal. On day 17, animals were either sacrificed after blood collection or subjected to ischemia–reperfusion (IR) injury (30 min of coronary occlusion followed by 180 min or 24 h of reperfusion for IS or serum cTnI level determination, respectively). The colon was removed and colon length was measured, then the different anatomical sections of the GI tract were stored in 10% buffered formalin until use for histological analysis.

**Figure 3 biomedicines-11-02945-f003:**
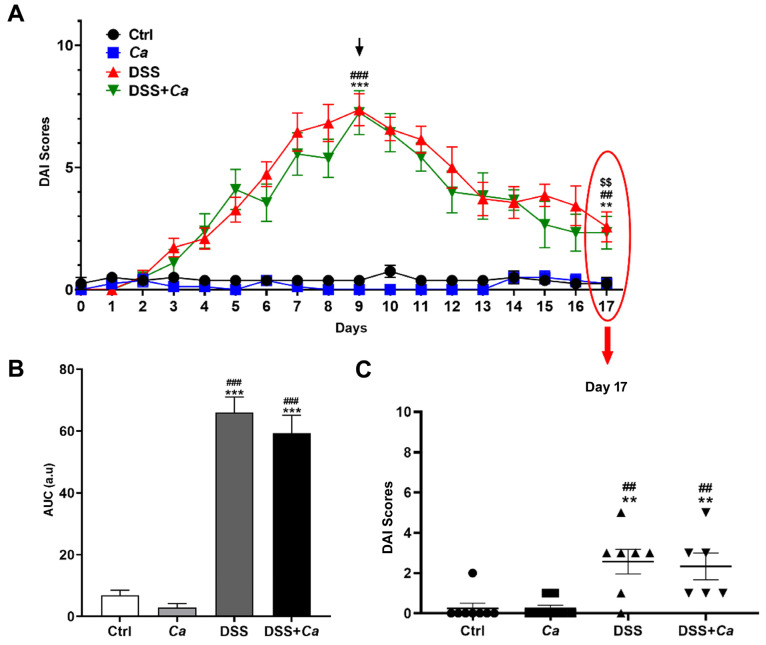
Disease activity index evolution. Animals were monitored daily for weight loss, stool consistency, and bleeding. (**A**) Disease activity index (DAI) scores were attributed based on body weight loss, fecal consistency, and hematochezia. DSS administration was stopped on day 9 (black arrow) and replaced with regular water and disease evolution continued to be monitored until sacrifice of animals on day 17 (highlighted with red oval, linking time point at 17 days with panel 3C, red arrow). (**B**) DAI area under the curve (AUC) analysis showed a significant difference between DSS-treated and control groups. (**C**) DAI on day 17 showing significant remission of IBD mice. **, *p* < 0.01 and ***, *p* < 0.001 vs. control (Ctrl) group; ##, *p* < 0.01, ###, *p* < 0.001 vs. *C. albicans* group (*Ca*); $$, *p* < 0.01 vs. corresponding group on day 9.

**Figure 4 biomedicines-11-02945-f004:**
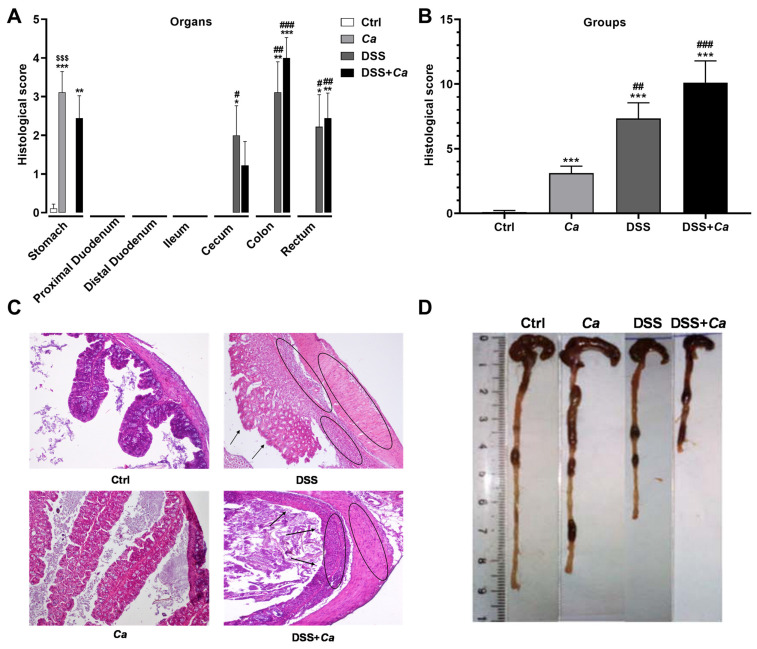
Histological analysis of the gastrointestinal tract in DSS-treated animals. (**A**) Histopathology scores were determined in each organ or section of the gastrointestinal tract. (**B**) Overall gastrointestinal tract inflammation score was assigned to each experimental group. (**C**) Representative H&E-stained colon section (magnification 40×) images. (**D**) Representative images of the colon on day 17. Colon length was measured in the different experimental groups. *, *p* < 0.05, **, *p* < 0.01, ***, *p* < 0.001 vs. control (Ctrl) group; #, *p* < 0.05, ##, *p* < 0.01, ###, *p* < 0.001 vs. *Ca* group. $$$, *p* < 0.001, vs. corresponding DSS group.

**Figure 5 biomedicines-11-02945-f005:**
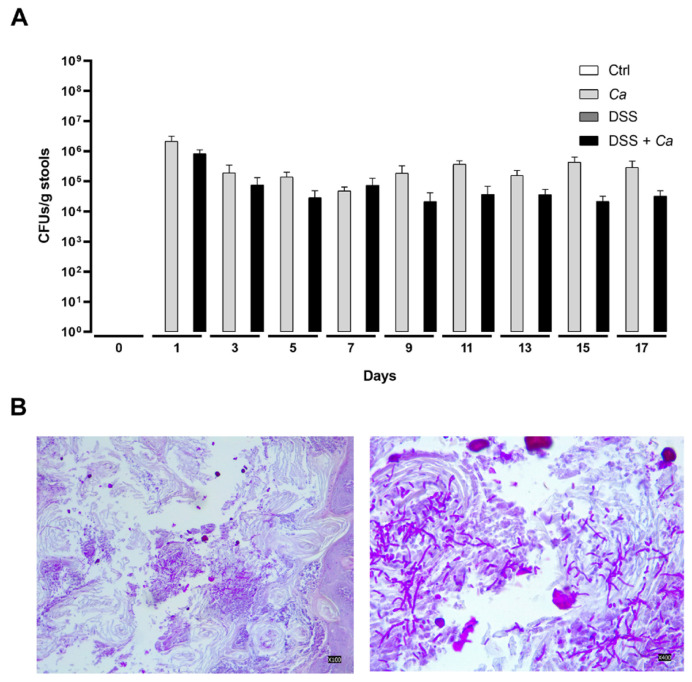
*Candida albicans* colonization analysis. (**A**) Quantification of *C. albicans* throughout the duration of the experiment in the different groups. Stool samples were homogenized in PBS and then plated onto solid YPD medium in Petri dishes and incubated at 30 °C for 48 h. *Ca* colonies were counted and colonization was determined as colony-forming units per gram of stool (CFU/g). (**B**) Representative PAS-stained colon sections (magnification 10×, left panel; and 40×, right panel) images showing invasions and colonization of GI walls by *C. albicans* with different morphologies (yeast and hyphae).

**Figure 6 biomedicines-11-02945-f006:**
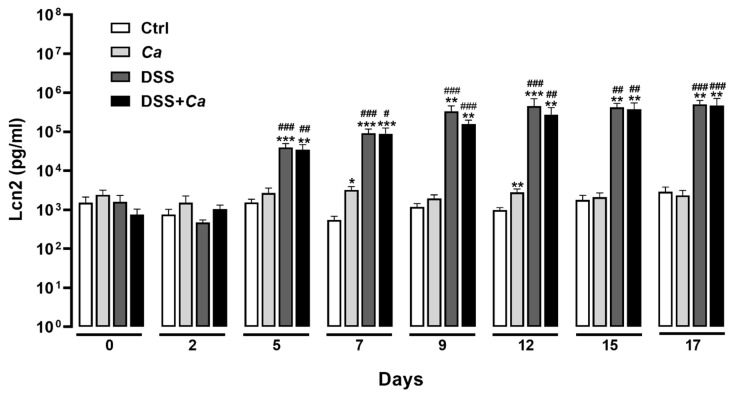
Local inflammatory marker fecal lipocalin 2 (Lcn2) analysis by ELISA assay. Tests were performed on suspensions of stool samples diluted to 1/10 for the control and *Ca* groups and to 1/50 and 1/100 for the DSS groups. The graph was generated as the log_10_ of the concentration of each group on the day of sampling. *, *p* < 0.05, **, *p* < 0.01, ***, *p* < 0.001 vs. corresponding control (Ctrl) group; #, *p* < 0.05, ##, *p* < 0.01, ###, *p* < 0.001 vs. corresponding *Ca* group.

**Figure 7 biomedicines-11-02945-f007:**
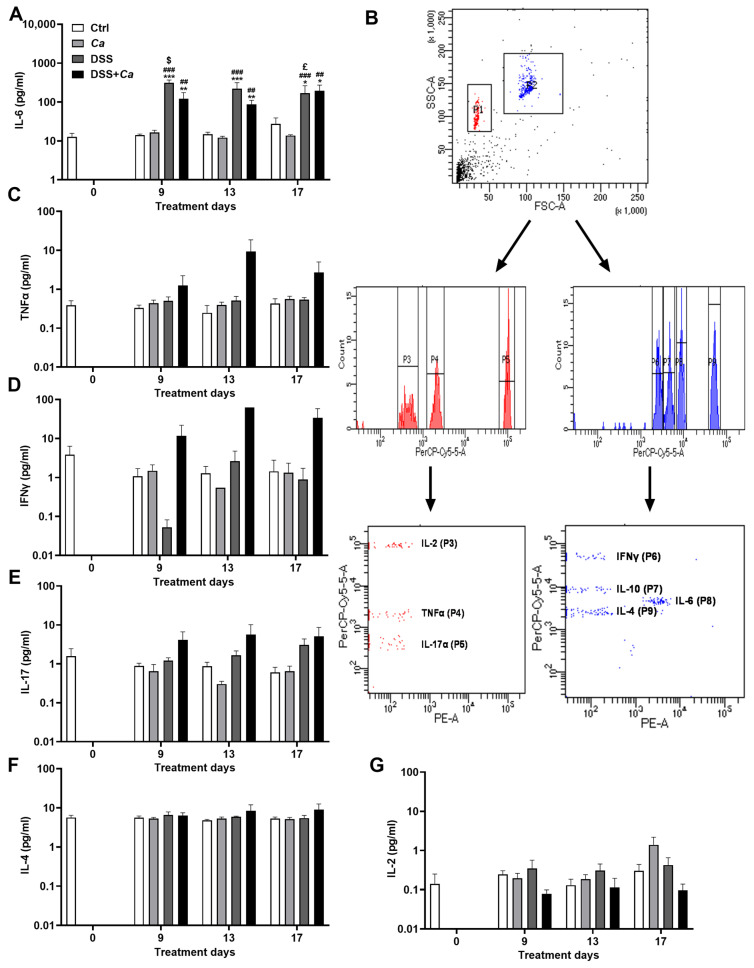
Systemic inflammatory cytokine analysis by flow cytometry. Graphs were generated as log_10_ of each group’s concentration. (**A**) Interleukin-6 (IL-6) expression levels in the different experimental groups. (**B**) Representative flow cytometry plots showing the variable cytokine distribution in DSS-treated group. (**C**) Tumor necrosis factor alpha (TNF-α) expression levels in the different experimental groups. (**D**) Interferon gamma (IFNγ) expression levels in the different experimental groups. (**E**) Interleukin-17 (IL-17) expression levels in the different experimental groups. (**F**) Interleukin-4 (IL-4) expression levels in the different experimental groups. (**G**) Interleukin-2 (IL-2) expression levels in the different experimental groups. *, *p* < 0.05, **, *p* < 0.01, ***, *p* < 0.001 vs. corresponding control (Ctrl) group; ##, *p* < 0.01, ###, *p* < 0.001 vs. corresponding *C. albicans (Ca*) group*;* $, *p* < 0.05 vs. corresponding DSS group; £, *p* < 0.05 vs. corresponding group on day 9.

**Figure 8 biomedicines-11-02945-f008:**
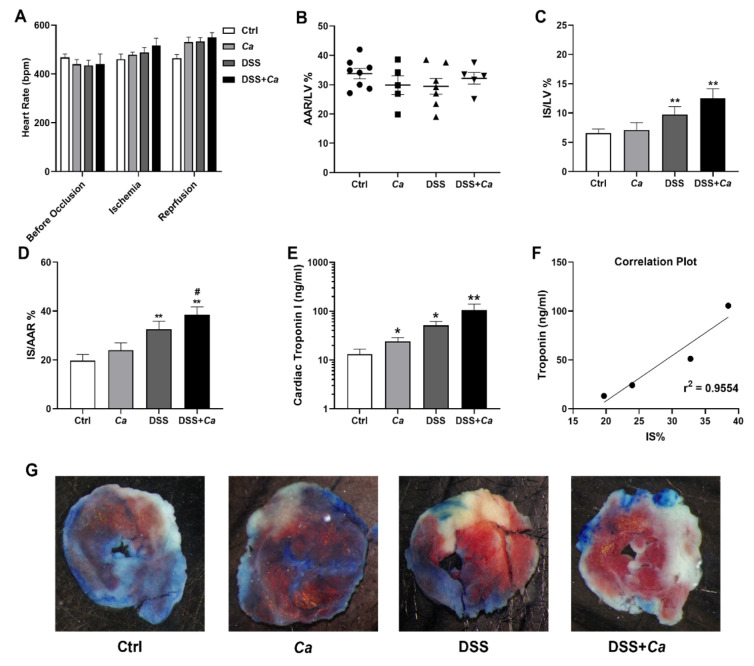
Assessment of IR severity in IBD mice. IR was determined by TTC staining expressed as percentage of area at risk (IS/AAR) or cardiac troponin I (cTnI) serum level quantification by ELISA, after 30 min ischemia followed by 3 or 24-h reperfusion, respectively. (**A**) Heart rate (HR) before coronary occlusion, during ischemia, and after reperfusion in the different experimental groups. (**B**) The area at risk (AAR) was determined after 30 min of ischemia followed by 3 h of reperfusion by intravenous injection of Evans blue solution. (**C**) Assessment of infarct size (IS) as a percentage of the left ventricle (IS/LV %). (**D**) Assessment of IS as a percentage of AAR (IS/AAR %). (**E**) Measurement of cTnI serum levels in mice with or without treatment (DSS/*Ca*). The graph was generated as the log_10_ of the concentration of each group. (**F**) Correlation between IR (IS/AR) and cTnI serum levels in the four experimental groups after IR (*r*^2^ = 0.9554, *p* < 0.05). (**G**) Planimetry of myocardial IS. Digital photographs of midventricular slices after Evans blue and triphenyl tetrazolium chloride (TTC) staining. Infarcted areas appear pale, viable myocardium within the AAR is stained brick red. *, *p* < 0.05, **, *p* < 0.01 vs. control (Ctrl) group; #, *p* < 0.05 vs. *C. albicans (Ca*) group.

**Figure 9 biomedicines-11-02945-f009:**
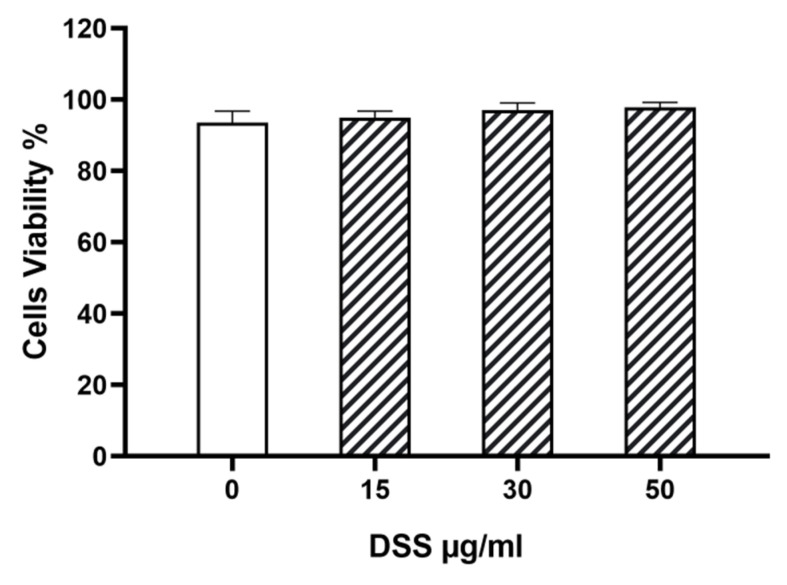
DSS effect on cardiomyocyte viability in vitro. Human induced pluripotent stem cell (HiPSC)-derived cardiomyocytes (HiPSC-CMs) were incubated with DSS at increasing concentrations (15, 30, and 50 µg/mL) for 48 h and cell viability was assessed by flow cytometry. Cells were tested in triplicate and results expressed as mean ± SEM for each concentration.

**Figure 10 biomedicines-11-02945-f010:**
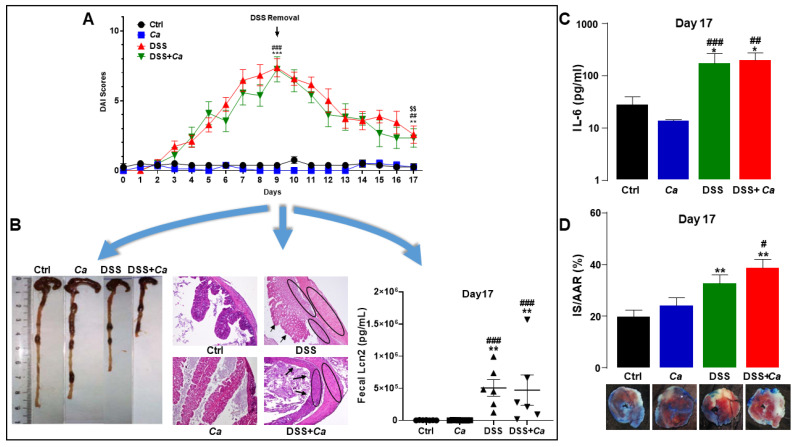
IBD–MI relationship model in mice. (**A**) Colitis in mice was induced by oral treatment with 1% (*w*/*v*) DSS for 9 days followed by an 8-day remission period, as evidenced by disease activity index (DAI). On day 17, myocardial ischemia–reperfusion (IR) was performed in IBD mice, while (**B**) local [histology (magnification 40×), colon length, and fecal lipocalin 2 (Lcn2)] and (**C**) systemic (IL-6) inflammation remained higher than in the control group, allowing (**D**) the study of the impact of IBD on the severity of MI (IS/AAR%). *, *p* < 0.05, **, *p* < 0.01, ***, *p* < 0.001 vs. corresponding control (Ctrl) group; #, *p* < 0.05, ##, *p* < 0.01, ###, *p* < 0.001 vs. corresponding Ca group; $$, *p* < 0.01 vs. corresponding group on day 9.

**Table 1 biomedicines-11-02945-t001:** Disease activity index scoring system based on body weight loss, fecal consistency, and hematochezia.

Weight Loss *		Stool Consistency		Blood in Stools	
Observation	Score	Observation	Score	Observation	Score
≤2%	0	Normal	0	No blood	0
3–6%	1	Semi-solid	1	Gross blood in stools	2
7–12%	2	Loose to pasty	2	Rectal bleeding	4
>12%	3	Diarrhea	3		

* Changes in body weight are indicated as percentage of body weight loss reported to baseline at day 0.

**Table 2 biomedicines-11-02945-t002:** Histological score attribution system.

		Inflammatory Cell Infiltration	Intestinal Architecture
SCORE	0	Rare inflammatory cells in the *lamina propria*	Absence of lesions on the mucosa
	1	Increased number of inflammatory cells, including neutrophils in the *lamina propria*	Discrete focal lymphoepithelial lesions
	2	Confluence of inflammatory cells, with extension to the submucosa	Mucosal erosion/ulceration
	3	Transmural extension of inflammatory cell infiltrate	Extensive mucosal damage and extension to deep structures of the intestinal wall

**Table 3 biomedicines-11-02945-t003:** Mean values ± SEM for body weight, left ventricular weight to body weight ratio, and area at risk in IBD mice subjected to ischemia–reperfusion in basal condition or after DSS, *C. albicans* (*Ca*), or DSS + *Ca* treatments.

					Heart Rate (bpm)		
Groups	n	BW (g)	LVW/BW (%)	AAR/LV (%)	Before Ischemia	In Ischemia	In Reperfusion
Ctrl	8	27.9 ± 1.5	3.3 ± 0.9	33.8 ± 1.8	468 ± 14	461 ± 21	464 ± 16
DSS	7	27.0 ± 0.9	3.8 ± 0.5	29.4 ± 2.7	435 ± 21	488 ± 21	533 ± 16
*Ca*	5	28.3 ± 0.6	3.5 ± 0.04	29.9 ± 3.2	440 ± 20	478 ± 11	531 ± 20
DSS + *Ca*	5	28.5 ± 0.5	3.3 ± 0.11	32.2 ± 0.2	441 ± 41	517 ± 30	550 ± 19

Data are means ± SEM. BW, body weight; LVW/BW, left ventricle weight to body weight ratio; AAR, area at risk; HR, heart rate; bpm, beats per minute.

## Data Availability

All data generated or analyzed during this study are included in this published article.

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
