# Peer review of "Inflammatory Bowel Disease Increases the Severity of Myocardial Infarction after Acute Ischemia–Reperfusion Injury in Mice"

_biomedicines, 2023, doi:10.3390/biomedicines11112945_

Round 1
Reviewer 1 Report
The manuscript entitled, “Inflammatory bowel disease increases the severity of myocardial infarction after acute ischemia-reperfusion injury in mice" by Mami et al., investigated the the relationship between inflammatory bowel disease (IBD) and myocardial infarction (MI). Inflammatory bowel disease model was created by administering dextran sulfate sodium (DSS) in drinking water alone or with oral gavage of Candida albicans. Clinical, histological, and biomarker assessments of intestinal and systemic inflammation were used to determine inflammatory bowel disease severity. The experiment tested myocardial infarction severity in mice by ischemia-reperfusion (IR). This was done by assessing the infarct size and blood cTnI levels. The mice with inflammatory bowel disease (IBD) demonstrated increased concentrations of lipocalin 2 (Lcn2) and interleukin-6 (IL-6) in their fecal samples. The DSS mice demonstrated a nearly twofold increase in infarct size (IS) compared to the control group. Additionally, there was a substantial correlation observed between blood cardiac troponin I (cTnI) levels and infarct size. The administration of Ca inoculation was found to exacerbate systemic inflammation and injury to insulin resistance (IR) generated by DSS. However, it is important to note that these findings did not reach statistical significance. This work serves as an initial proof-of-concept, providing evidence of the influence of inflammatory bowel disease (IBD) on the severity of myocardial infarction (MI).
Although the article under review has commendable writing quality, significance, and timeliness, there are certain modifications that this reviewer would suggest in order to provide a more comprehensive analysis of the subject matter.
Comments:
1, The English of the manuscript could be refined (minor), and there are few typographical errors.
2, Authors can include the limitations to their study.
3, In order to emphasize the study's summary, at least one graphical illustration may be included.
4, Authors should add few citations about immune cells such as adaptive immune cell’s role during Myocardial Infarction (PMID: 36382968, PMID: 36093172, PMID: 31898271, PMID: 37115472, PMID: 36337927 etc).
Minor editing of English language required
Author Response
Response to Reviewer 1 Comments
Comments and Suggestions for Authors
The manuscript entitled, “Inflammatory bowel disease increases the severity of myocardial infarction after acute ischemia-reperfusion injury in mice" by Mami et al., investigated the the relationship between inflammatory bowel disease (IBD) and myocardial infarction (MI). Inflammatory bowel disease model was created by administering dextran sulfate sodium (DSS) in drinking water alone or with oral gavage of Candida albicans. Clinical, histological, and biomarker assessments of intestinal and systemic inflammation were used to determine inflammatory bowel disease severity. The experiment tested myocardial infarction severity in mice by ischemia-reperfusion (IR). This was done by assessing the infarct size and blood cTnI levels. The mice with inflammatory bowel disease (IBD) demonstrated increased concentrations of lipocalin 2 (Lcn2) and interleukin-6 (IL-6) in their fecal samples. The DSS mice demonstrated a nearly twofold increase in infarct size (IS) compared to the control group. Additionally, there was a substantial correlation observed between blood cardiac troponin I (cTnI) levels and infarct size. The administration of Ca inoculation was found to exacerbate systemic inflammation and injury to insulin resistance (IR) generated by DSS. However, it is important to note that these findings did not reach statistical significance. This work serves as an initial proof-of-concept, providing evidence of the influence of inflammatory bowel disease (IBD) on the severity of myocardial infarction (MI).
Although the article under review has commendable writing quality, significance, and timeliness, there are certain modifications that this reviewer would suggest in order to provide a more comprehensive analysis of the subject matter.
Comments (in black) and answers (in red):
Point 1: The English of the manuscript could be refined (minor), and there are few typographical errors.
Response 1: First, we would like to thank the reviewer for the time spent reading and editing this document, as well as for the insightful comments to improve the quality of this article. The manuscript text was revised to correct typographical errors, orthography as well as grammatical and sentence structures. All changes made are now included in the revised version of the manuscript.
Point 2: Authors can include the limitations to their study.
Response 2: We thank reviewer #1 for this insightful comment. Indeed, our work presents limitations such as the lack of confirmation of the mechanistic causal link between IBD and the severity of MI. The first objective of our experimental work was to respond, through animal experimentation, to an unresolved question which is whether IBD could increase the severity of infarction in patients with MI. In this work, we address this question by setting up a mouse model combining IBD and MI. This experimental work required intensive and lengthy preliminary experiments (time, mice, reagents, surgery, different experimental procedures, etc) to achieve the current results. Numerous optimizations were carried out to determine the best dose of DSS to use in mice (5%, 4%, 3%, 2.5%, 2%, 1.5% and 1%) as well as the treatment period (5, 7, 9 and 17 days), so that IBD is compatible with animal survival after ischemia-reperfusion (IR) experiments. Despite all these optimizations, IR resulted in up to 50% losses in the IBD animals tested for this work. As IR is a very delicate procedure involving coronary occlusion and cardiac surgery on small live rodents, animal mortality was one of the limiting factors in our work, especially during the reperfusion phase. For these reasons, we were unable to further explore in vivo the mechanistic links between IBD and MI, particularly those related to IBD-induced IL-6, which is suggested to be the cause of increased MI severity in our experiments. However, we are currently considering developing other approaches (such as an in vitro model) to address in the future this issue and further explore the mechanistic link between IBD and MI, especially the role of IL-6 in the observed cardiac effects.
Limitations of the study (text below) have been included in the “Discussion” section (just before the” Conclusion” paragraph, page 36, line 638-651) of the revised version of the manuscript:
Although our results provide novel insights into the interaction between IBD and MI, they leave some unanswered questions. Indeed, we demonstrated for the first time that IBD significantly aggravated the severity of MI and that these effects could involve IL-6 pathway activation. However, the role of IL-6 as a causal link between IBD and MI was not further explored in this work. Several reports have demonstrated that, in both IBD and MI, inhibition of IL-6 can prevent disease development, even in patients refractory to clinical trials with anti-TNF-α therapies [79, 96, 97]. Thus, IL-6 can be a potential biomarker for MI prognosis and a target to improve the treatment of IBD patients. In the future, further studies will be needed to assess the role of this cytokine in the predisposition to CVD in individuals with IBD. Furthermore, in our work, microbiota modification by acute inoculation of C. albicans had no impact on DSS induced-colitis, suggesting that multiple C. albicans administrations will likely be required for a higher impact on MI. Additionally, further investigations will be needed to determine whether other gut microflora products, such as metabolites, are responsible for the severity of IS in the DSS-induced colitis model.
Point 3: In order to emphasize the study's summary, at least one graphical illustration may be included.
Response 3: We thank the reviewer for this comment. To avoid redundancies in the article, we have not added other figures. However, we have modified Figure 10 which is already included in the “Discussion” section as a summary of the main conclusions of the work, in order to simplify the understanding of the results. Additionally, Figures 1 and 2 in the “Materials and Methods” section summarize the experimental protocol and overall study design, respectively.
Point 4: Authors should add few citations about immune cells such as adaptive immune cell’s role during Myocardial Infarction (PMID: 36382968, PMID: 36093172, PMID: 31898271, PMID: 37115472, PMID: 36337927 etc).
Response 4:
We thank reviewer #1 for this comment. We added a paragraph in the “Discussion” section discussing the role of immune cells during myocardial infarction using the references indicated in the comment (lines 581-602 in the revised version).

Reviewer 2 Report
The author studied the relationship between myocardial infarction (MI) and inflammatory bowel disease (IBD), they used the DSS-induced IBD model for this study, and found that IBD mice exhibited high level of Lcn2 and IL-6, which aggravated the myocardial infarction. Also they indicated the oral C. albicans treatment has no effect on MI during DSS-induced IBD model through statistical analysis. Overall, this study suggests that IBD aggravates MI. However, I think there are a few key issues that need to be clarified and addressed before publication.
1 Although the authors demonstrate that IBD aggravates MI, are there other measures to alleviate IBD to see if the symptoms are alleviated? Only in this way can we finally prove the conclusion of the study.
2 The authors found that IL-6 is crucial, could the author observe the effect of reversing MI after intervention with IL-6 antibody or something? Phenomena are not telling.
3 The Fig 10 is a little bit strange, I suggest the author should revise this figure.
4 At the same time, the references are too old, even 10 years ago, and it is recommended to update them.
Author Response
Response to Reviewer 2 Comments
Comments and Suggestions for Authors
The author studied the relationship between myocardial infarction (MI) and inflammatory bowel disease (IBD), they used the DSS-induced IBD model for this study, and found that IBD mice exhibited high level of Lcn2 and IL-6, which aggravated the myocardial infarction. Also they indicated the oral C. albicans treatment has no effect on MI during DSS-induced IBD model through statistical analysis. Overall, this study suggests that IBD aggravates MI. However, I think there are a few key issues that need to be clarified and addressed before publication.
Comments (in black) and answers (in red):
Point 1: Although the authors demonstrate that IBD aggravates MI, are there other measures to alleviate IBD to see if the symptoms are alleviated? Only in this way can we finally prove the conclusion of the study.
Response 1: First of all, we would like to thank reviewer #2 for the time spent evaluating our work and for the valuable comments. We hope that our answers and clarifications will help improve the quality of our manuscript. We set up an IBD model by oral administration of DSS (at 1% for 9 days, followed by a remission phase of 8 days without treatment). This model resulted in an aggravation of MI by inducing a twofold increase in infarct size (IS) compared to non-IBD mice. Prior to this model (DSS, 1%, 9 days), we performed several preliminary studies and optimizations using DSS at different doses and durations (4, 3.5 and 2.5%), all of which induced high mortality in mice within the first days of the protocol (before subjecting them to ischemia-reperfusion (IR) experiments). The model presented in our article is therefore the one using the lowest DSS dose (1%) that can be tested in our case, which unfortunately does not allow to answer to the excellent comment raised by reviewer #2 (i.e. alleviating IBD and determine if MI severity decreases as well). In the literature, IBD models are performed with DSS doses up to 5% with a treatment period longer than 9 days. However, these patterns could not be reproduced in our mouse model. These discrepancies with the literature may be linked to the sensitivity of the mouse strain used (in our case C57BL/6 mice versus BALBc mice in the literature). Additionally, the deleterious effect of DSS on gastrointestinal tract may differ depending on the batch used and the supplier (as recommended by the supplier). However, in our IR experiments, we noticed that mice developing severe IBD were less resistant to IR and died before the end of reperfusion, suggesting that IBD severity influences susceptibility to cardiac ischemia. Furthermore, the quantitative determination of blood cardiac troponin I, 24 hours after IR, confirmed our results obtained by cardiac planimetry analyses. Finally, in experiments performed in vitro on cardiomyocytes, we verified that DSS had no deleterious effects per se on the cells, suggesting that the cardiac effects observed in DSS mice are directly related to IBD disease.
Point 2: The authors found that IL-6 is crucial, could the author observe the effect of reversing MI after intervention with IL-6 antibody or something? Phenomena are not telling.
Response 2: In our study, serum IL-6 showed increased levels in IBD mice among a panel of seven cytokines tested, which makes it a potential target for a therapeutic approach to alleviate the severity of MI. However, the role of IL-6 as a major causal link between IBD and MI severity has not yet been investigated in our study. The primary objective of the current work was to determine whether IBD could impact ischemia tolerance using animal experimentation, in order to provide an answer to a discordant human hypothesis for which observational studies could not provide any direct evidence. However, we expect to further explore the role of IL-6 in IBD-induced cardiac effects through the use of IL-6 inhibitors. In the literature there is already a growing body of evidence that, in both IBD and MI diseases, inhibition of IL-6 prevents disease development, even in patients refractory to clinical trials with other therapies such as anti-TNF-α. We have already considered exploring the role of Il-6 (and other cytokines) in the IBD/MI interaction, but we were slowed down by the very significant mortality of IBD mice after IR (up to 50%). Therefore, we are currently studying other experimental approaches to further explore the mechanistic role of IL-6 in the observed cardiac effects.
The response to this comment (Point 2) is now addressed in the “Discussion” section (just before the” Conclusion” paragraph, page 36, line 638-651) of the revised version of the manuscript:
Although our results provide novel insights into the interaction between IBD and MI, they leave some unanswered questions. Indeed, we demonstrated for the first time that IBD significantly aggravated the severity of MI and that these effects could involve IL-6 pathway activation. However, the role of IL-6 as a causal link between IBD and MI was not further explored in this work. Several reports have demonstrated that, in both IBD and MI, inhibition of IL-6 can prevent disease development, even in patients refractory to clinical trials with anti-TNF-α therapies [79, 96, 97]. Thus, IL-6 can be a potential biomarker for MI prognosis and a target to improve the treatment of IBD patients. In the future, further studies will be needed to assess the role of this cytokine in the predisposition to CVD in individuals with IBD. Furthermore, in our work, microbiota modification by acute inoculation of C. albicans had no impact on DSS induced-colitis, suggesting that multiple C. albicans administrations will likely be required for a higher impact on MI. Additionally, further investigations will be needed to determine whether other gut microflora products, such as metabolites, are responsible for the severity of IS in the DSS-induced colitis model.
Point 3: The Fig 10 is a little bit strange, I suggest the author should revise this figure.
Response 3: We have introduced some changes in the design of Figure 10 (and its legend) which represents a summary of the study results.
Point 4: At the same time, the references are too old, even 10 years ago, and it is recommended to update them.
Response 4: We thank reviewer #2 for this comment. We have revised the article references and updated them as much as possible. Some references are old because they are original articles relevant to our work. Furthermore, studies that focus on the relationship between the two diseases (IBD and MI), are mainly retrospective case studies without much information on animal models.
